# Effect of antiretroviral therapy on decreasing arterial stiffness, metabolic profile, vascular and systemic inflammatory cytokines in treatment-naïve HIV: A one-year prospective study

**Pedro Martínez-Ayala**[1], **Guillermo Adrian Alanis-Sánchez**[2], **Monserrat Álvarez-Zavala**[3], **Karina Sánchez-Reyes**[3], **Vida Verónica Ruiz-Herrera**[1], **Rodolfo Ismael Cabrera-Silva**[3], **Luz Alicia González-Hernández**[1,3], **Carlos Ramos-Becerra**[2], **Ernesto Cardona-Muñoz**[2], **Jaime Federico Andrade-Villanueva**[1,3]*

1 HIV Unit Department, University Hospital "Fray Antonio Alcalde", University of Guadalajara, Guadalajara, Mexico, 2 Department of Physiology, Arterial Stiffness Laboratory, University of Guadalajara, Guadalajara, Mexico, 3 Clinical Medicine Department, HIV and Immunodeficiencies Research Institute, CUCS-University of Guadalajara, Guadalajara, Mexico

* hhaller@live.com.mx

## Abstract

### Introduction

Cardiovascular disease is a major cause of death among people living with HIV (PLH). Non-treated PLH show increased levels of inflammation and biomarkers of vascular activation, and arterial stiffness as a prognostic cardiovascular disease risk factor. We investigated the effect of one year of ART on treatment-*naïve* HIV(+) individuals on arterial stiffness and inflammatory and vascular cytokines.

### Methods

We cross-sectionally compared aortic stiffness via tonometry, inflammatory, and vascular serum cytokines on treatment-*naïve* (n = 20) and HIV (-) (n = 9) matched by age, sex, metabolic profile, and Framingham score. We subsequently followed young, treatment-*naïve* individuals after 1-year of ART and compared aortic stiffness, metabolic profile, and inflammatory and vascular serum biomarkers to baseline. Inflammatory biomarkers included: hs-CRP, D-Dimer, SAA, sCD163s, MCP-1, IL-8, IL-18, MRP8/14. Vascular cytokines included: myoglobin, NGAL, MPO, Cystatin C, ICAM-1, VCAM-1, and MMP9.

### Results

Treatment-*naïve* individuals were 34.8 years old, mostly males (95%), and with high smoking prevalence (70%). Baseline T CD4$^+$ was 512±324 cells/mcL. cfPWV was similar between HIV(-) and treatment-*naïve* (6.8 vs 7.3 m/s; p = 0.16) but significantly decreased after ART (-0.52 m/s; 95% CI -0.87 to -0.16; p0.006). Almost all the determined cytokines were significantly higher compared to controls, except for MCP-1, myoglobin, NGAL,

**Data Availability Statement:** Data are available from Harvard Database at https://dataverse.harvard.edu/dataset.xhtml?persistentId=doi:10.7910/DVN/MGQGKG.

**Funding:** The author(s) received no specific funding for this work.

**Competing interests:** The authors have declared that no competing interests exist.

cystatin C, and MMP-9. At follow-up, only total cholesterol and triglycerides increased and all inflammatory cytokines significantly decreased. Regarding vascular cytokines, MPO, ICAM-1, and VCAM-1 showed a reduction. D-Dimer tended to decrease (p = 0.06) and hs-CRP did not show a significant reduction (p = 0.17).

## Conclusion

One year of ART had a positive effect on reducing inflammatory and vascular cytokines and arterial stiffness.

## Introduction

Antiretroviral treatment (ART) has become effective in controlling infections caused by human immunodeficiency virus (HIV). Worldwide, by 2021, the United Nations has estimated that approximately 37.7 million people are currently living with human HIV. In 2020, 73% of patients with HIV had access to ART [1]. In Mexico, until May 2021, 113,788 HIV patients were registered as active for ART [2]. HIV infection itself is associated with the development of cardiovascular disease (CVD) [3,4], atherosclerosis [5], and arterial stiffness [6]. Cardiovascular complications are among the leading causes of morbidity and mortality in patients with HIV. ART and HIV infection have a complex interaction with inflammation, coagulation, and other factors. Higher levels of envelope glycoprotein 120, Nef protein, interleukin-6, high-sensitivity C-reactive protein (hs-CRP), and D-dimer have been associated with endothelial dysfunction, cardiovascular disease, and all-cause mortality [7,8]. In addition, HIV infection has been linked with immune activation and low-grade chronic inflammation [9]. It has been reported that traditional CVD scores (i.e., Framingham Risk Score and Atherosclerotic Cardiovascular Disease Risk Score) systematically underestimate cardiovascular risk in HIV [10].

Aortic arterial stiffness is a predictor of cardiovascular events independent of traditional risk factors [11]. Carotid-femoral pulse wave velocity (cfPWV) is considered the gold standard for measuring aortic stiffness [12]. A recent meta-analysis including 17 studies investigating HIV and arterial stiffness found an overall increased cfPWV in individuals with HIV (+0.44 m/s) [13]. Furthermore, some studies have reported adverse effects of ART on aortic stiffness [14,15], whereas others did not report any association [16,17]. A possible explanation for these discrepancies may be the different combinations of ART regimens, different populations, and methods used to assess arterial stiffness. Most studies on arterial stiffness and HIV are cross-sectional and cannot establish a causal relationship; unfortunately, there is a lack of longitudinal clinical studies that investigate the effect of ART and cfPWV over time with the evaluation of inflammatory and vascular cytokines, especially in developing countries. We have previously reported increased arterial stiffness in treatment-*naïve* HIV individuals compared to HIV(-) controls [18]. The underlying mechanisms linking HIV infection with arterial stiffness remain unclear. Therefore, the present study aimed to investigate the effect of one year of ART on arterial stiffness, inflammatory and metabolic serum biomarkers in treatment-*naïve* HIV individuals.

## Materials and methods

### Study population

Between January 2015 and August 2019, people living with HIV (PLH) were enrolled at the "Antiguo Hospital Civil de Guadalajara" in Guadalajara, Mexico. The study complied with the

Declaration of Helsinki and was approved by the ethics committee of the Hospital Civil Fray Antonio Alcalde (Register number: 208/15). After approval from the ethics committee, all individuals who attended the HIV Unit to start ART were invited to participate in the study. Informed consent was obtained from all participants. At study entry, the participants' medical history and demographic information were obtained using a questionnaire. Inclusion criteria for PLH included: a) Patients 18 years of age or older with confirmed HIV infection and no previous ART, b) Absence of current or previous rheumatological or neoplastic disease or CVD, c) Not taking any vasoactive medication (e.g., antihypertensives, vasopressors, etc.), d) Without opportunistic infections at the time of enrolment. As a pilot prospective study, we analyzed the first 20 PLH from our previous cross-sectional study [18] that achieved and sustained virologic suppression for one year and who had complete data. To avoid selection bias, we confirmed that the demographic and clinical characteristics did not differ from those of the remaining 31 individuals from our cohort, which were not included in the analysis (S1 Table). The control group was paired by age and sex and recruited from our local network of researchers and volunteers within the University of Guadalajara. In addition, general laboratory testing and medical interrogation were performed on control individuals to confirm similar lifestyle as possible (except for the smoking habit which is higher in most HIV cohorts worldwide [19]) and metabolic and cardiovascular risk profiles. Inclusion criteria for the control group included: a) Negative HIV serological test, b) No previous cardiovascular, metabolic, or rheumatological disease, c) Not on any medication, including vasoactive drugs.

## Biomarkers

**Arterial stiffness.** Arterial stiffness was measured by cfPWV as described previously [12] by applanation tonometry (PulsePen, Diatechne, Milan, Italy). cfPWV was calculated as the time delay between the arrival of the pulse wave at the carotid and the femoral artery, divided by the distance between the carotid and femoral arteries, with previous automated subtraction of the segment between the carotid and the sternal notch by the software. All measurements were performed by a single trained technician in a temperature-controlled room. The participants rested in a supine position for 15 minutes before the assessment and were instructed to abstain from smoking, and alcoholic, or caffeinated beverages 12 hours before the evaluation [20]. Systolic (SBP) and diastolic blood pressure (DBP) were measured using an automated sphygmomanometer (Omron HEM-907XL). Mean arterial pressure (MAP) was calculated as MAP = DBP + peripheral pulse pressure (pPP)'0.33.

**Viremic control and immunological status.** A venous blood sample in an EDTA-tube was obtained from the antecubital vein after 8-hour fasting. $CD4^+$ T-cells count was determined by flow cytometry (FACScalibur System, Becton Dickinson) and HIV-1 viral load was determined using with real-time polymerase chain reaction with retro transcription (Cobas AmpliPrep/Cobas Taqman, Roche Diagnostics) in a federal laboratory.

**Metabolic profile.** Serum samples were obtained from the antecubital vein after an 8-hour fasting. Serum lipids, including total cholesterol (TC), high-density lipoprotein cholesterol (HDL-c), low-density lipoprotein cholesterol (LDL-c), and triglycerides (TG), were determined by colorimetric quantification (AU5800 autoanalyzer, Coulter Beckman, USA). Plasma glucose was determined by photometry (AU5800 autoanalyzer, Beckman Coulter, USA) in a central laboratory.

**Cytokine and vascular inflammation proteins quantification by flow cytometry.** Three bead-based multiplex assays were employed to quantify cytokines and vascular biomarkers in all participants: LEGENDplex[TM] Human Inflammation Panel, LEGENDplex™ Human Vascular Inflammation Panel 12P, and finally detection of hs-CRP was performed with the

LEGENDplex™ Human Vascular Inflammation Panel 1S/P Plex (BioLegend, Inc., San Diego, CA, USA). All assays were performed in accordance with the manufacturer's instructions. Data were acquired in an Attune Acoustic Focusing Cytometer (Life Technologies, Carlsbad, CA, USA). More than 2,000 events for each analyte were acquired. The files were analyzed using LEGENDplex$^{TM}$ Data Analysis Software. Values are expressed in pg/mL.

**D-Dimer and sCD163 quantification.** Serum samples were immediately stored at -80°C. Quantification of hs-CRP, D-dimer, and sCD163 was performed using D-Dimer Human SimpleStep and sCD163 (M130) Human, both by ELISA (Abcam®) following the manufacturer's instructions. Values are expressed as ng/mL.

## Statistical analyses

Depending on their distribution, data are presented as mean± standard deviation and median [interquartile range]. Parametric and non-parametric tests for independent and paired variables were used accordingly. Chi-square was used for non-paired and McNemar´s test for paired categorical data. Statistical significance was set at a two-tailed p-value of <0.05. Cliff's delta is a non-parametric effect size estimate that was calculated to assess the effect of ART on inflammatory and vascular cytokines [21]. Effect size thresholds for Cliff's delta were <0.147 for negligible, 0.148–0.33 for small, 0.334–0.474 for medium, and 0.475 for large effects. Based on previous data from our lab, we calculated a Pearson correlation coefficient between cfPWV paired data of 0.81 and the sample size necessary to detect a paired difference of 0.6 m/s and pooled SD 1.1, which resulted in a sample size of 12 individuals with 80% power. In addition, the effect size of ART on IL-10 after a 12-month ART was 5.1 requiring 6 patients in total [22], and 0.96 for VCAM, requiring 19 patients in total [23]. Statistical analysis and graphical representations were performed with GraphPad Prism 6. Cliff's delta was calculated using RStudio v.1.3.1073 (Vienna, Austria) with the package "effsize" (v0.8.0, Torchiano, 2020) [24]. Sample size calculation was performed using G*Power V.3.1.9.6 (Universität Kiel, Germany). Due to the explorative nature of this study and the small sample size, a correlation analysis was not performed. Future studies with sufficient statistical power to establish an association between cytokine levels and arterial stiffness changes are required.

## Results

### Demographic data

Patient´s characteristics at enrolment are shown in Table 1. Age and sex distributions were similar between the HIV(-) and treatment-*naïve* HIV(+) groups, with a higher prevalence of males in both groups. ART was initiated within one month of enrolment, fourteen patients (73.6%) with EFV/TDF/FTC, two patients (10.5%) with ATV/r/TDF/FTC, and three patients (15.7%) with DVR/r/TDF/FTC. Among modifiable CVD risk factors, there was a significantly higher proportion of active smokers in the treatment-*naïve* HIV(+) group, with no changes in smoking at follow-up. BMI was similar between HIV(-) and treatment-*naïve* HIV(+) groups and did not change after one year of ART.

### Metabolic data

Regarding the lipid profile, the treatment-*naïve* HIV(+) group exhibited higher TG (p = 0.003) and lower c-HDL (p<0.01) compared to HIV(-), with no significant changes at one-year follow-up. Although TC and c-LDL levels were similar between groups at baseline, in the treatment-*naïve* HIV(+) group, TC significantly increased (p = 0.011) and c-LDL showed a trend toward a significant increase post-ART (p = 0.055). Plasma glucose levels were not significantly

**Table 1. Demographic, metabolic, HIV and vascular data from HIV negative and treatment-*naïve* HIV individuals at baseline and 1-year post ART follow-up (Post-ART).**

| | HIV (-)<br>(n = 9) | HIV (+) | |
| --- | --- | --- | --- |
| | | *Naïve*<br>(n = 20) | Post-ART<br>(n = 20) |
| **Demographics** | | | |
| Age, years | 34.4 ± 8.2 | 34.8 ± 10.1 | - |
| Male sex, % | 8 (88.8) | 19 (95) | - |
| Cigarette smoking, % | 1 (11.1) | 14 (70)* | 14 (70) |
| BMI, kg/m$^2$ | 24.3 ± 1.8 | 25.3 ± 4.1 | 25.9 ± 4.2 |
| **Metabolic profile** | | | |
| Glucose, mg/dL | 87.5 ± 5.5 | 84.4 ± 8.7 | 85.4 ± 7.2 |
| TC, mg/dL | 185.6 ± 41.0 | 167.0 ± 41.5 | 185.6 ± 44.6† |
| c-LDL, mg/dL | 116.7 ± 37.1 | 100.2 ± 32.5 | 112.8 ± 38.3 |
| c-HDL, mg/dL | 50.2 ± 10.3 | 33.7 ± 9.4* | 35.3 ± 8.5 |
| TG, mg/dL | 93.0 ± 40.3 | 167.0 ± 67.5* | 176.3 ± 72.7* |
| Framingham score, % | 2.8 [1–3.3] | 3.3 (1.9–7.9) | 3.3 (2.3–7.0) |
| **HIV status** | | | |
| T CD4$^+$, cells/mcL | - | 512 ± 324 | 727 ± 306† |
| T CD8$^+$, cells/mcL | - | 977 (733–11374) | 859 (572–1180) |
| CD4$^+$/CD8$^+$ ratio | - | 0.39 (0.27–0.44) | 0.73 (0.39–1.48) † |
| Viral load ≥50 cop/mL, n (%) | - | 20 (100) | 0 (0) †† |
| **Hemodynamics** | | | |
| SBP, mmHg | 111 ± 11 | 108 ± 13 | 116 ± 12† |
| DBP, mmHg | 66 ± 8 | 67 ± 8 | 67 ± 10 |
| MAP, mmHg | 81 ± 7 | 80 ± 8 | 83 ± 10 |
| pPP, mmHg | 45 ± 14 | 39 ± 11 | 49 ± 11† |
| cfPWV, median, m/s | 6.8 (5.8–7.7) | 7.3 (6.7–8.1) | 6.8 (5.8–7.8) † |

Values are presented as median (IQR) and mean ± SD. BMI, body mass index; TC, total cholesterol; c-LDL, low-density lipoprotein cholesterol; c-HDL, high-density lipoprotein cholesterol; TG, triglycerides; T CD4$^+$, lymphocytes T CD4; T CD8$^+$, lymphocytes T CD8$^+$; SBP, systolic blood pressure; DBP, diastolic blood pressure; MAP, mean arterial pressure; pPP, peripheral pulse pressure; cfPWV, carotid-femoral pulse wave velocity. *p<0.05 *vs* HIV(-), † *vs* HIV(+) *naïve*. *p<0.01 *vs* HIV(-), †p<0.01 *vs* HIV *naïve*.

different between HIV(-) and treatment-*naïve* HIV(+) and did not change with ART. Both HIV(-) and treatment-*naïve* HIV(+) exhibited similar Framingham scores (p = 0.145), with no changes post-ART (p = 0.400).

### HIV variables

All treatment-*naïve* HIV(+) individuals achieved virologic suppression (viral load <50 copies/ mL). CD4$^+$ T-cell count increased by 36% from baseline (p = 0.003), CD8$^+$ T-cells did not change (p = 0.14), but the CD4$^+$/CD8$^+$ ratio increased at follow-up (p<0.001).

### Blood pressure and arterial stiffness

The treatment-*naïve* HIV(+) group presented similar SBP, DBP, MAP, and pPP at baseline (all p>0.05), compared to HIV(-), with a significant increase only in SBP (p = 0.04) and pPP (p = 0.005) at 1-year follow-up. Baseline cfPWV was similar between treatment-*naïve* HIV(+) and HIV(-) (p = 0.16); however, the post-ART group showed a significant reduction (-0.52 m/ s; 95% CI -0.87 to -0.16; p = 0.006) (Fig 1).

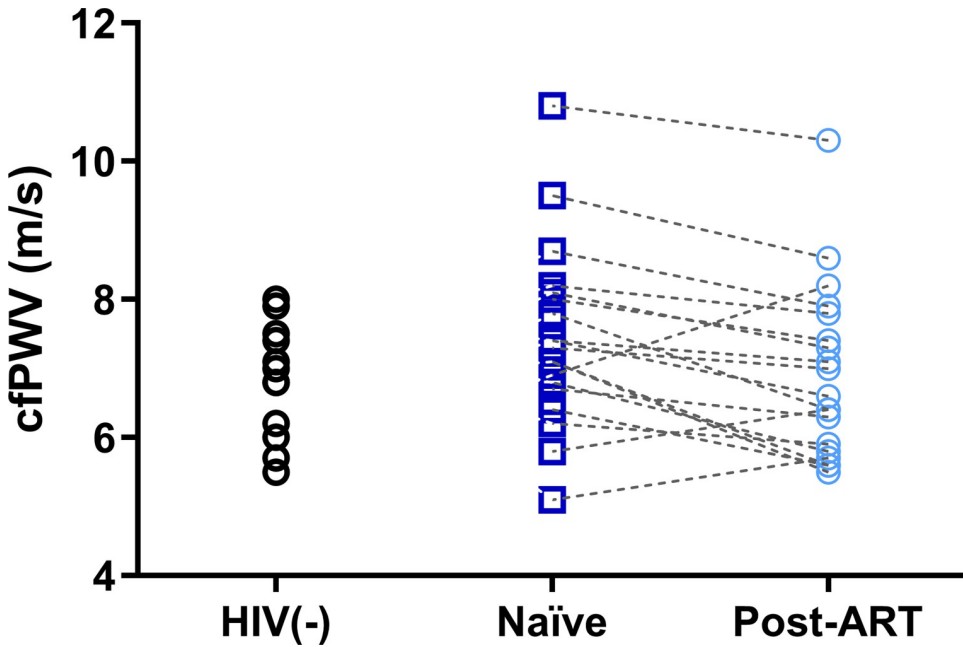

**Fig 1. Carotid-femoral pulse wave velocity (cfPWV) in HIV(-), treatment-*naïve* HIV(+), and after one-year follow-up of antiretroviral therapy (post-ART).**

## Inflammation Biomarkers

Inflammation biomarkers are shown in Table 2. At baseline, SAA, sCD163, IL-8, IL-18, calprotectin (i.e., MRP8/14), hs-CRP, and D-dimer levels were higher in treatment-*naïve* HIV(+) compared to HIV(-) (all p<0.05), while MCP-1 levels were similar (p = 0.247). After ART, most biomarkers of inflammation significantly decreased (p<0.01), except for D-dimer which showed a trend toward a significant decrease (p = 0.063). Similarly, hs-CRP (p = 0.187) was reduced to almost half its baseline value, but the difference was not statistically significant. Despite ART, hs-CRP levels remained significantly higher in the post-ART group than in the control group. The effect size of ART on inflammation biomarkers is shown in Fig 2A. Cliff's delta showed a medium effect on SAA (-0.38; 95% CI -0.10 to -0.65) and a large on sCD163 (-0.63, 95% CI -0.27 to -0.83), IL-8 (-0.62; 95% CI 0.27 to 0.82), IL-18 (-0.54; 95% CI -0.16 to -0.77), calprotectin (-0.52; 95% CI -0.16 to -0.76), and MCP-1 (-0.53; 95% CI -0.16 to -0.77).

## Vascular inflammation biomarkers

The vascular inflammation biomarkers are shown in Table 2. Treatment-*naïve* HIV(+) had higher D-dimer, ICAM-1, VCAM-1, and MPO levels, compared to HIV(-) (all p<0.05), while neutrophil gelatinase-associated lipocalin (NGAL) tended to be higher (p = 0.08). Myoglobin, Cystatin C, and MMP-9 levels were similar between the treatment-*naïve* HIV(+) and HIV(-) groups. At follow-up, treatment-*naïve* HIV(+) showed a significant reduction in ICAM-1, VCAM-1, neutrophil gelatinase-associated lipocalin (NGAL), and MPO. On the other hand, MPO, ICAM-1, and VCAM1 were not significantly different between the post-ART and HIV (-), except for NGAL which was lower than HIV(-) (p = 0.029). The effect size of ART on vascular inflammation biomarkers is shown in Fig 2B. Cliff´s delta effect size showed a large effect on NGAL (-0.66; 95% CI -0.30 to -0.85), MPO (-0.65; 95% CI -0.29 to -0.85), and VCAM (-0.78; 95% CI -0.45 to -0.92).

**Table 2. Inflammatory, vascular, and metabolic cytokines in controls, HIV+ _naïve_ and HIV+ treated.**

| Cytokines | HIV (-) (n = 9) | HIV (+) | |
|---|---|---|---|
| | | _Naïve_ (n = 20) | Post-ART (n = 20) |
| **Inflammatory** | | | |
| SAA, ng/mL | 143.8 (84.9–445.5) | 916.7 (440.5–1714)* | 307.6 (212.6–996.9)† |
| sCD163s, ng/mL | 40683 (34261–45137) | 81701 (54210–116553)* | 45768 (40090–75773)† |
| MCP-1, pg/mL | 812.1 (608.8–1178) | 972.2 (781.8–1471) | 653.8 (362.4–793.7)† |
| IL-8, pg/mL | 4.9 (3.2–36.8) | 189.1 (68.8–519.1)* | 12.2 (3.1–127.9)† |
| IL-18, pg/mL | 171.4 (74.1–296.4) | 357.4 (166.1–707.5)* | 99.6 (82.0–237.8)† |
| MRP8/14, ng/mL | 4569 (2182–7312) | 6988 (5214–11153)* | 4398 (2939–5593)† |
| hs-CRP, mg/dL | 0.4 (0.2–0.6) | 2.8 (0.6–6.6)* | 1.1 (0.7–2.2)* |
| D-Dimer, ng/mL | 225 (27–370) | 694 (211–1886)* | 366 (205–570) |
| **Vascular** | | | |
| Myoglobin, ng/mL | 20.6 (14.7–48.2) | 30.5 (22.2–44.0) | 24.5 (13.5–49.2) |
| NGAL, ng/mL | 261.5 (107.0–380.2) | 381.1 (240.7–569) | 5.8 (2.5–197.8)*† |
| MPO, ng/mL | 45.1 (12.5–107.7) | 132.6 (91.5–233.8)* | 45.1 (8.1–82.2)† |
| Cystatin C, mg/mL | 0.22 ± 0.09 | 0.25 ± 0.10 | 0.20 ± 0.10 |
| ICAM-1, ng/mL | 3.14 (1.9–4.0) | 6.56 (5.5–8.1)* | 4.3 (1.9–6.6)† |
| VCAM-1, ng/mL | 1107 (1040–1172) | 2541 (2067–2881)* | 791.6 (17.8–1530)† |
| MMP-9, ng/mL | 563.6 ± 295.7 | 693.6 ± 408.7 | 681 ± 379.6 |

Values are presented as median (IQR) and mean ± SD. SAA, serum amyloid A; sCD163, soluble CD163; MCP-1, monocyte chemoattractant protein-1; IL-8, interleukin 8; IL-18; interleukin 18; MRP8/14, myeloid-related protein 8/14 calprotectin; hs-CRP, high-sensitivity C-reactive protein; NGAL, neutrophil gelatinase-associated lipocalin; MPO, myeloperoxidase; ICAM, intercellular adhesion molecule 1; VCAM, vascular cell adhesion molecule 1; MMP-9, matrix metalloproteinase-9. *p<0.05 _vs_ HIV(-), † _vs_ HIV(+) treatment-_naïve_.

## Discussion

In this study, we investigated the effects of one-year ART on arterial stiffness and inflammatory and vascular cytokines levels in non-elderly, treatment-_naïve_ PLH. First, we observed that the treatment-_naïve_ HIV(+) group presented higher levels of inflammatory, vascular cytokines, and arterial stiffness compared to HIV(-) controls. Second, after a one-year follow-up, cfPWV and some cytokines significantly decreased from baseline levels (before ART) and reached levels similar to those in the control group. However, hs-CRP remained higher post-ART, which may suggest a persistent low-grade inflammatory state despite ART.

As previously reported by Smith et al., [25] we observed a higher smoking prevalence among the HIV population (70%). We also found that the treatment-_naïve_ HIV(+) group presented the typical dyslipidemia pattern in PLH, characterized by high TG and low HDL levels [26]. In our population, the most common treatment was EFV/TDF/FTC (70%), which showed an unfavorable lipid profile outcome, as previously reported by Daar et al. [27].

The arterial system acts as a conduit for blood and as a buffering system to deliver a steady flow to key organs such as the brain and kidneys. Their buffering capacity depends on how elastic/stiff arteries are; in this sense, arterial elasticity is influenced by the vasomotor tone and structural characteristics of the arterial wall (elastin and collagen content). The vasomotor tone is regulated by the endothelium, sympathetic tone, and vasoactive hormones such as the renin-angiotensin-aldosterone system (RAAS). Nitric oxide (NO) is a key molecule responsible for regulating vasomotor tone. During inflammation, released cytokines can induce endothelial dysfunction by limiting the availability of tetrahydrobiopterin, the precursor of NO [28], and decrease the capacity of endothelial NO synthase (eNOS) to produce NO [29],

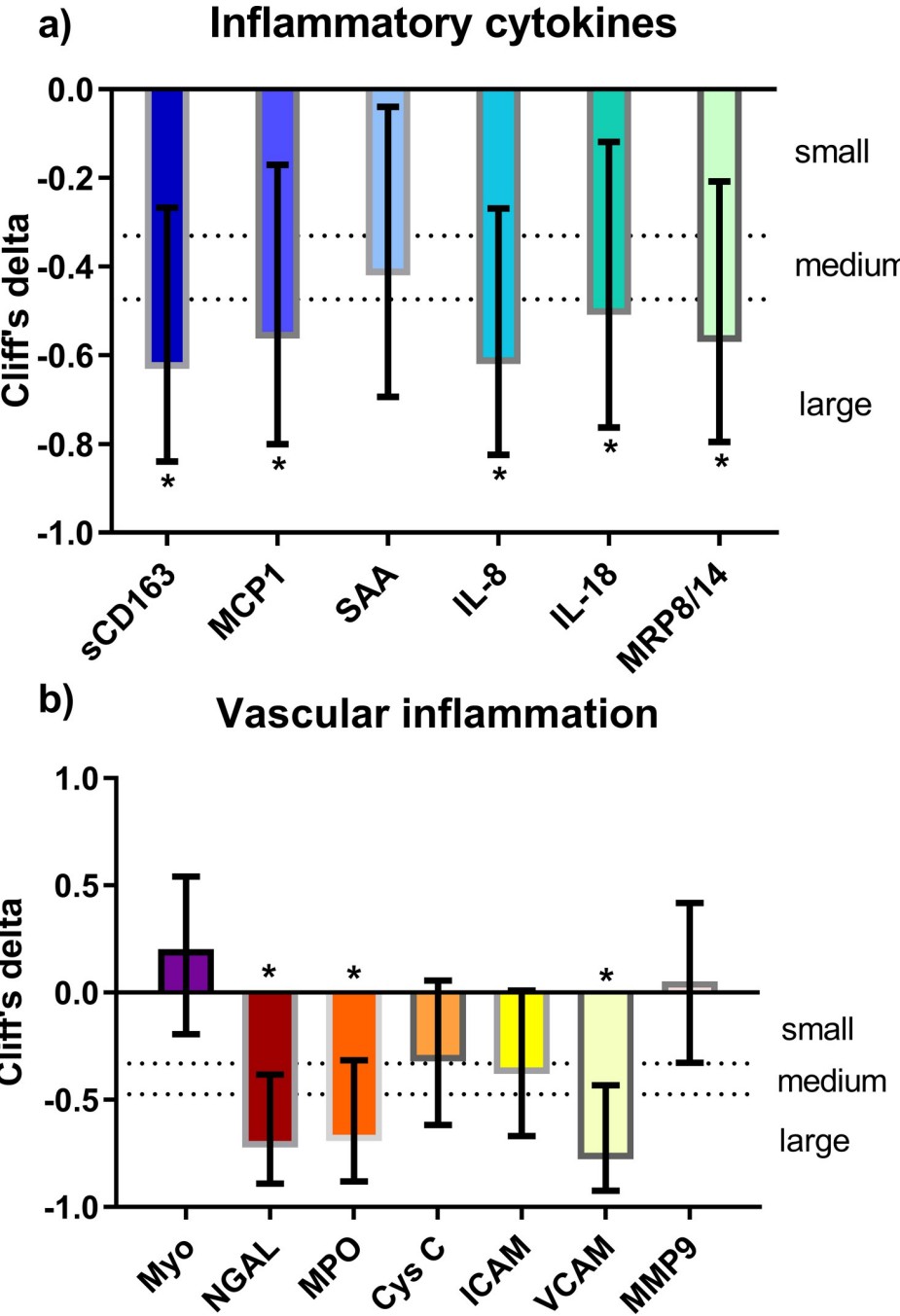

**Fig 2.** Non-parametric effect size (Cliff's delta) and 95% confidence interval (error bars) of the effect of ART on a) inflammatory cytokines and b) markers of vascular inflammation. sCD163, soluble CD163; MCP-1, monocyte chemoattractant protein-1; SAA, serum amyloid A; IL-8, interleukin 8; IL-18; interleukin 18; MRP8/14, myeloid-related protein 8/14 (calprotectin); NGAL, neutrophil gelatinase-associated lipocalin; MPO, myeloperoxidase; Cys C, cystatin C; ICAM, intercellular adhesion molecule VCAM, vascular cell adhesion molecule 1; MMP-9, matrix metalloproteinase-9. *p<0.05 significant reduction compared to baseline.

subsequently limiting the capacity to reduce vasomotor tone. In HIV, several factors can affect the vasculature, including endothelial dysfunction [30], RAAS hyperactivation, infection of vascular smooth muscle cells, increased coagulation, chronic immune activation (by the virus

itself and by microbial translocation caused by enteropathy), abnormal cholesterol metabolism, lipoprotein transportation, and platelet activation [31,32]. In our study, we found a significant decrease in several inflammatory and vascular cytokines, a halving of hs-CRP levels, and a decrease in PWV in the post-ART group. Our findings differ from those of Rose et al., [33] who did not observe a decrease in PWV after 1-year of ART. Conversely, Maia-Leite et al., [34] reported that in their ART-experienced group, 87.9% of individuals virologically suppressed that aortic stiffness was similar to HIV(-) controls, implying the importance of viral suppression in arterial health. Our findings suggest that there is an improvement in arterial function, based on lower inflammation and a better environment for recovering normal arterial function. Nevertheless, we consider that future studies with complementary arterial evaluation techniques (e.g., flow mediation dilation) could provide better insights into the mechanisms responsible for recovering normal arterial function in inflammatory states.

The data obtained by measuring the vascular inflammation biomarkers in our study were consistent with previously reported data. Teasdale et al., [35] reported a significant decrease in D-dimer levels after 6 months of ART; [36,37]. In our study, we observed an overall trend toward a significant decrease in D-dimer post-ART. However, patients treated with ART based on transcriptase inhibitors (both non-nucleoside analogue and nucleoside analogue) presented a reduction (79%) in D-dimer levels, while patients with ritonavir-boosted PI-based ART D-dimer increased by 1.35 times. These data are consistent with the association of ritonavir-boosted PI exposure time with CVD risk and suggest that it can appear as soon as a year of ritonavir-boosted PI-based ART [36].

In the present study, MPO concentrations decreased after ART and could be associated with a PWV improvement. El-Bejanni et al. [38] reported a negative correlation between MPO and CD4[+] T-cell counts. Thus, immune reconstitution in these individuals is accompanied by a decrease in MPO and PWV. MPO and aortic stiffness have been related to plaque instability in atherosclerotic disease [37,39,40]. Therefore, it could be considered that the global effect of one year on ART is reflected by an improvement in arterial health and probably less CVD risk.

Another interesting vascular marker is NGAL, which is associated with inflammation, leukocyte migration, carotid stenosis, endothelial dysfunction, plaque formation, acute myocardial infarction, and chronic heart failure [41,42]. It has been demonstrated that NGAL can be expressed in macrophages, smooth muscle cells, and endothelial cells; moreover, it can activate the NF-κB pathway, promoting the expression of several proinflammatory cytokines such as IL-8, MCP-1, TNF-α, and IL-1β [43]. We found a significant decrease in NGAL levels, which could be proposed as a novel cardiovascular biomarker in PLWH.

ICAM-1 and VCAM-1 have been associated with atherosclerosis [44] and CVD risk even in HIV(-) population [30,35,37,38]. In our study, we observed a substantial decrease in the levels of VCAM-1 after ART, as reported by other groups [45], reaching similar levels to the HIV(-) population, as well as ICAM-1 which showed a reduction although non statistically significant. MPO and aortic stiffness have been related to plaque instability in atherosclerotic disease [37,39,40]. In the present study, MPO concentrations also decreased after ART and could be associated with a PWV improvement. El-Bejanni et al. [38] reported a negative correlation between MPO and CD4[+] T-cell counts. Thus, the immune reconstitution of these individuals is accompanied by a decrease in both variables; therefore, it could be considered that the global effect of one year on ART is reflected in an improvement in arterial health and probably less CVD risk.

It has been reported that PLH exacerbates the production of proinflammatory cytokines, such as IL-1β, IL-8, and TNF-α, and molecules related to non-canonical activation pathways of coagulation, such as IL-6, hs-CRP, SSA, and D-dimer [38,46–49]. In line with those studies, we observed that the concentrations of SAA, sCD163, MCP-1, IL-8, and IL-18 were increased

in PLH, but after a year of treatment, most HIV (+) patients reached concentrations similar to controls, except for SAA and sCD163. These results suggest that ART reduced the degree of systemic inflammation and improved arterial stiffness after one year of treatment.

It is well described that ART does not completely suppress viral load, particularly in the viral reservoirs. Hence, low-grade chronic inflammation persists and contributes to an increased risk of non-AIDS-defining events such as CVD [50,51]. As we previously mentioned, there was a reduction of SAA, sCD163, and calprotectin after a year of successful ART; however, the levels remained higher compared to controls. SAA, like CRP, is an acute-phase protein produced by the liver. It is known that SAA can establish an inflammatory atherosclerotic and thrombotic microenvironment that impacts immune dysfunction and promotes CVD [52,53]. In our study, we observed that SAA was higher in treatment-*naïve* compared to controls and reduced post-ART to levels similar to those in HIV(-). High concentrations of calprotectin were detected before ART; however, serum levels normalized after the virologic control, this pattern is consistent with previous data reported since calprotectin is one of the first proteins to increase in plasma, even earlier than other markers of myocardial necrosis. Finally, calprotectin promotes atherosclerosis, and elevated plasma concentrations of calprotectin are considered predictors of future CV events [42,43,46].

As PLH live longer, new widely available biomarkers are necessary to properly evaluate the overall health of this population and to prepare physicians for the more plausible complications their patients may have. This study has some prospectively analyzed biomarkers that could be of use; however, they require further investigation to evaluate their long-term effectiveness in the daily clinical field.

These results are consistent with the idea that PLH develops a low-grade inflammation state and metabolic imbalance from the beginning of HIV infection, which remains despite ART. Furthermore, it seems that besides the current therapy goals (i.e., CD4$^+$ T-cell count and HIV RNA levels) it would be advisable to monitor arterial health with different available techniques to try to lower the CV risk in this population, which increases with age and ART exposure.

## Strengths and limitations

This study has some limitations. Despite our aim to isolate the effect of HIV on arterial stiffness by choosing PLH patients without any other comorbidities and comparing them with healthy controls, there may be a series of genetic, lifestyle, social, and behavioral factors that differ between the populations studied, which could not be controlled for. Our study sample included mostly males, and sex differences could play a role in the response of arterial stiffness and levels of inflammatory and vascular biomarkers to ART. Another limitation of the present study was the absence of cytokine reference values for the Mexican population to determine whether HIV-infected individuals achieved normal cytokine levels after ART. The sample size of the control group was smaller than that of the HIV(+) group, which may have been underpowered for some biomarkers. Lastly, we only measured PWV twice (baseline and at 12 months); thus, we could not determine how far into the treatment the PWV and cytokines started to decrease. Nevertheless, the strength of this study is the prospective evaluation of arterial health and serum biomarkers of inflammation which allowed us to study the effects of ART.

## Conclusion

One year of ART had a positive effect on reducing both inflammatory and vascular cytokines, and arterial stiffness.

## Supporting information

**S1 Table. Demographic and clinical characteristics of patients included and not included in the final analysis.**
(DOCX)

## Acknowledgments

This article is dedicated to the memory of Dr. Silvia G. Esquivel Razo, whose kindness and passion for science and service were an inspiration for all of us. Her legacy will remain in our hearts and minds.

## Author Contributions

**Conceptualization:** Pedro Martínez-Ayala, Guillermo Adrian Alanis-Sánchez, Luz Alicia González-Hernández, Carlos Ramos-Becerra, Ernesto Cardona-Muñoz, Jaime Federico Andrade-Villanueva.

**Data curation:** Guillermo Adrian Alanis-Sánchez, Monserrat Álvarez-Zavala, Karina Sánchez-Reyes, Rodolfo Ismael Cabrera-Silva, Luz Alicia González-Hernández.

**Formal analysis:** Pedro Martínez-Ayala, Guillermo Adrian Alanis-Sánchez, Vida Verónica Ruiz-Herrera.

**Methodology:** Guillermo Adrian Alanis-Sánchez, Carlos Ramos-Becerra, Ernesto Cardona-Muñoz.

**Writing – original draft:** Pedro Martínez-Ayala, Guillermo Adrian Alanis-Sánchez, Vida Verónica Ruiz-Herrera.

**Writing – review & editing:** Pedro Martínez-Ayala, Monserrat Álvarez-Zavala, Karina Sánchez-Reyes, Rodolfo Ismael Cabrera-Silva, Luz Alicia González-Hernández, Ernesto Cardona-Muñoz, Jaime Federico Andrade-Villanueva.

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
