## [Decision Letter · Decision Letter 0]

31 Oct 2022

PONE-D-22-26706Effect of antiretrovirals on the regression of arterial stiffness, metabolic, vascular, and systemic inflammatory cytokines at one year of virologic controlPLOS ONE

Dear Dr. Andrade-Villanueva, 

Thank you for submitting your manuscript to PLOS ONE. After careful consideration, we feel that it has merit but does not fully meet PLOS ONE’s publication criteria as it currently stands. Therefore, we invite you to submit a revised version of the manuscript that addresses the points raised during the review process.

Dear Dr. Andrade,

Thank you for submit your manuscript to PLOS One. Both reviewers are experts in the area and I hope you and your group appreciate the constructive and positive comments to your work. Please make the suggested changes if you agreed. If you do not agreed please contact me to discussed with the group.

Best Regards,

Eliseo Eugenin

Please submit your revised manuscript by Dec 15 2022 11:59PM. If you will need more time than this to complete your revisions, please reply to this message or contact the journal office at plosone@plos.org. Please include the following items when submitting your revised manuscript:

We look forward to receiving your revised manuscript.

Kind regards,

Eliseo A Eugenin, Ph.D.

Academic Editor

PLOS ONE

Journal Requirements:

Reviewers' comments:

Reviewer's Responses to Questions

**Comments to the Author**

1. Is the manuscript technically sound, and do the data support the conclusions?

Reviewer #1: Yes

Reviewer #2: Yes

2. Has the statistical analysis been performed appropriately and rigorously? 

Reviewer #1: Yes

Reviewer #2: Yes

3. Have the authors made all data underlying the findings in their manuscript fully available?

Reviewer #1: Yes

Reviewer #2: Yes

4. Is the manuscript presented in an intelligible fashion and written in standard English?

Reviewer #1: Yes

Reviewer #2: Yes

5. Review Comments to the Author

Reviewer #1: Effect of antiretrovirals on the regression of arterial stiffness, metabolic, vascular, and systemic inflammatory cytokines at one year of virologic control

Pedro Martínez-Ayala PlosOne

HIV creates an increase inflammation with vascular consequences and been only partially reversed by ART. Such inflammation is driven by HIV itself and also by leaky gut allowing microbial translocation of bacterial and fungus products.

Investigators assessed the effect of one-year treatment-naïve HIV individuals on arterial stiffness and inflammatory and vascular cytokines. tonometry, inflammatory, and vascular serum cytokines on treatment-naïve (n=20) and HIV uninfected (n=9) age matched control with metabolic profile, and Framingham score evaluation.

Study findings indicate that ART had a significant effect on reducing inflammatory and most vascular cytokines and arterial stiffness.

Comments:

As nearly half of participants receive and NNRT vs PI comparison of these 2 groups will be of interest as PI increased cholesterol and have a cumulative risk for CV events.

The difficulty of such studies with many markers only a few fit with the study hypothesis. Here the 2 inflammatory markers linked with non-AIDS events d-dimers and IL-6 were not different. Discussion on these 2 validated markers for CVV risks should be discussed in detail.

LPS, sCD14 and beta-d-glucan play an important role in HIV-related inflammation and are linked with CV risk by imaging. Measurement of these markers will add value to the manuscript

Isnard S, et al. Circulating β-d-Glucan as a Marker of Subclinical Coronary Plaque in Antiretroviral Therapy-Treated People With Human Immunodeficiency Virus. Open Forum Infect Dis. 2021 Mar 7;8(6):ofab109.

Isnard S, et al. Gut Leakage of Fungal-Related Products: Turning Up the Heat for HIV Infection. Front Immunol. 2021 Apr 12;12:656414.

In addition, the stress cytokine GDF-15 emerges as one of the best markers in CV disorders: ischemic conditions, atrial fibrillation and cardiac insufficiency and in 2022 in HIV. Assessment of one the best CV markers will be welcome for this study.

Royston L, . Growth differentiation factor-15 as a biomarker of atherosclerotic coronary plaque: Value in people living with and without HIV. Front Cardiovasc Med. 2022 Aug 26;9:964650.

Discussion is much too long and should not be structured, no subchapters are necessary. Investigators should focus only on novelty as many data are confirming previous findings.

Increased in Cholesterol and TG are considered to augment as return to health at one year as generally patients gain 2 kg initially taken for HIV. Therefore, early changes are not linked to inflammation or PI effects on lipids after one year of ART.

Limitations: absence of cellular markers of adaptative and innate immunity.

As cells were not assessed a small paragraph on the role of adaptive immunity cells, such as CD4 cells are acknowledged to participate to CVD pathogenesis (Emeson et al., 1996; Zhou et al., 2005). CD4 cells were found infiltrated in atherosclerotic plaques (Saigusa et al., 2020). Recent evidence indicates that T helper 1 (Th1) cells have pro-atherogenic roles, whereas regulatory T cells (Tregs) can play a dual role being both anti-atherogenic or pro-atherogenic (George et al., 2012; Maganto-Garcia et al., 2011).

Similarly in HIV, innate immunity: Elevated frequency of circulating non-classical monocytes (CD14dimCD16++) (Gu et al., 1998) and the intermediate CD14+CD16+ monocyte counts were associated to subclinical atherosclerosis (Hanna et al., 2017) and the expression of CX3CR1 on CD16+ monocytes predicted carotid artery thickness (Westhorpe et al., 2014)

Reviewer #2: This is important research. The study provides novel, prospective data in the field of HIV-associated CVD and warrants publication, even considering the relatively small number of participants. I do however think the manuscript could be improved before publication.

Major Comments:

My biggest concern is how the group of 20 HIV infected persons were recruited. The authors mention prior work with 51 individuals. Were these persons included from the prior cohort or are the participants all new? How were they selected if previously enrolled? Was everyone that was recruited followed up fully and included? Attrition? This needs to be discussed that possible bias may be evaluated. If they were chosen based on viral suppression alone, comparison with a virally unsuppressed group will be valuable and should be strongly considered.

A significant limitation is the fact that the study essentially only included males. This should be discussed as a significant limitation as sex-differences were not evaluated and females are already significantly under-represented in CV research.

There were isolated delta (change over time) correlations mentioned , however, I miss a dedicated correlation analysis. Relevant correlations (or lack thereof) needs to be reported. Viral load, CD4 count, smoking, inflammatory markers, vascular markers and aortic stiffness?

Varying ways of referring to HIV infected persons and the HIV cohorts are used in manuscript. Decide on terminology and use this consistently.

NB: Mean CD4 count in table 1 is different to what is written in the abstract and the manuscript text.

The discussion section is too long and reads as quite meandering. Every section should have a clear message and build on the message of the article. The final message of the article should be clear, and I think the authors should work on the conclusion (both in the abstract and the manuscript) to clearly summarise what they think their findings mean.

Comments:

Introduction:

line 58-59: Unreferenced, inaccurate statement. CVD is not the leading cause of death in HIV. Rather say one of the leading causes of CVD.

line 65 onward: Needs rewrite to clarify. There are sweeping statements without references and should be avoided. Smoking (as the authors mention later) has been associated with HIV in certain studies. This sentence reads as contradictory.

line 68: reference please

line 69 onward: Tonometry-based cfPWV is frequently employed in clinical research, but is not considered to be the definitive method of PWV measurement as it is implied in the text. The statement in the manuscript should be corrected to only read: 'cfPWV is considered the gold standard of aortic stiffness measurement.' Consensus has not been reached as to the definitive method to measure cfPWV and various techniques are well described.

read

Rajzer, Marek W; Wojciechowska, Wiktoria; Klocek, Marek; Palka, Ilona; Brzozowska-Kiszka, Małgorzata; Kawecka-Jaszcz, Kalina (2008). Comparison of aortic pulse wave velocity measured by three techniques: Complior, SphygmoCor and Arteriograph. Journal of Hypertension, 26(10), 2001–2007. doi:10.1097/hjh.0b013e32830a4a25

line 77: Sparse work is available from low- and middle income countries, however, recent data have shown similar findings in sub-Saharan Africa. This may serve to strengthen your rationale for the research:

read

Robbertse PS, Doubell AF, Innes S, et al. Pulse wave velocity demonstrates increased aortic stiffness in newly diagnosed, antiretroviral naïve HIV infected adults: A case-control study. Medicine. 2022; 101:e29721.

line 113, 148 and other locations: Define all abbreviations with first use.

line 150: p-value of <0.05

line 153-155: This does not seem correct. Paired samples t-test with 80% power to detect a 0.6m/s difference and the quoted SD with alpha=0.05, calculates to 29 pairs on SPSS. This is more than double the number quoted in the text.

Results:

Were any of the patients of vasoactive mediation? I think this is a worthwhile to mention, even if no-one was on these medicines.

I find the BMI findings the inverse of what I would expect when compared to the controls. Any ideas why this may be the case in your study?

Not all abbreviations used in the tables are found in the legends. Please double check. Tables should be able to be able to be freestanding from the article and still be easy to interpret.

line 186: p=0.055 is not significant according to your study’s predefined level of significance. Rather state a trend toward significance for accuracy.

line 205: Again, consistent use of wording to refer to the study groups. Post-ART should rather be ART-group or ART-experienced group.

line 208: ‘In the beginning’. Do the authors mean at baseline?

Dedicated correlation analysis is an omission in my opinion as discussed earlier.

Discussion:

I would restructure the first paragraph to emphasise your own novel, longitudinal findings first. The mention of lack of longitudinal studies belong in the introduction. Rather state own strengths, than others weaknesses in this critical paragraph.

line 254: ‘slightly or did not decrease’. Reword this to communicate your findings clearly. Median hs-CRP essentially halved after ART. I would say something like ‘The hs-CRP showed a strong trend towards decrease, however, this did not reach statistical significance. hs-CRP, despite ART, remained significantly higher in ART-experienced persons when compared to the control group.’

line 265-270: This is all true. However, I think the reader would benefit from better packaging of these facts using our current theoretical framework of the factors that underpin cfPWV (especially in light of your young cohort, with likely negligible amounts of atherosclerotic disease). Pressure amplification by peripheral arterial tone. Vasomotor arterial tone is modulated by endothelial function, sympathetic tone, and the RAAS system. The manuscript would benefit from the incorporation of key concepts from the following articles (In general and in the setting of HIV). Use your excellent data to examine some of these concepts and what you think actually drive the increased (and decreased on ART) PWV mechanistically. Your study is explorative and should make the best use of your data.

read:

1. Cavalcante JL, Lima JAC, Redheuil A, et al. Aortic Stiffness. J Am Coll Cardiol. 2011;57:1511–22.

2. Robbertse PS, Doubell AF, Innes S, et al. Pulse wave velocity demonstrates increased aortic stiffness in newly diagnosed, antiretroviral naïve HIV infected adults: A case-control study. Medicine. 2022; 101:e29721.

line 319: reference

line 320: contrasting what? Use clear language.

line 367-368: I would be careful with these strong statements. Your findings do not quite support this and I suggest rewording this statement. I would focus on what you could show: that was that ART had an overall positive effect (decreased vascular pathology markers, decreased markers of inflammation, and decreased cfPWV). The observation that aortic stiffness decreased with ART is important, as this means (as you stated) that a reversible component of aortic stiffness remains and there is likely a window before this becomes irreversible (collagen deposition, degradation of elastin etc). When this window is, remains unknown.

Furthermore, if cfPWV is used a surrogate of CV risk, you demonstrated increased risk compared to HIV uninfected persons at baseline. As cfPWV decreased at one year on ART (in a small group of virally supressed individuals), I would see this as a relative decrease in CV risk. Yet another reason to give ART. The residual risk despite ART is still of concern, and as you mentioned stratification may be an issue, as people on ART have higher CVD compared to those without HIV.

Limitations:

The sample is small and explorative and should be stated in no uncertain terms. Selection of the cohort needs to be more detailed, as there is a perceived risk of inclusion bias.

Typo’s, spelling, and language: Various errors present. I would suggest involving a proof-reader before re-submission.

line 30. PWHIV

line 36. young, treatment-naïve. Not treatment-naïve young.

line 60: envelop

line 211: inflammation. rather biomarkers of inflammation

Minor comments for your discretion:

Title: I miss the word HIV, even though antiretrovirals imply this. I do not agree with the word "regression". Consider rewriting the title to refer to a decrease in aortic stiffness.

line 61: We generally refer to hs-CRP as high-sensitivity CRP, not highly sensitive CRP

line 62: remove increased

line 64: non-related should be unrelated

---

I look forward to the amended version of the manuscript and the inevitable publication of this important research.

6. PLOS authors have the option to publish the peer review history of their article (what does this mean?). If published, this will include your full peer review and any attached files.

Reviewer #1: **Yes: **Jean-Pierre Routy

Reviewer #2: No

---

## [Author Response · Author response to Decision Letter 0]

20 Jan 2023

RESPONSE TO REVIEWERS

We wish to thank the reviewers for their insightful and kind comments. After doing the necessary modifications, this manuscript improved substantially.

Reviewer #1: Effect of antiretrovirals on the regression of arterial stiffness, metabolic, vascular, and systemic inflammatory cytokines at one year of virologic control

Pedro Martínez-Ayala PlosOne

HIV creates an increase inflammation with vascular consequences and been only partially reversed by ART. Such inflammation is driven by HIV itself and also by leaky gut allowing microbial translocation of bacterial and fungus products.

Investigators assessed the effect of one-year treatment-naïve HIV individuals on arterial stiffness and inflammatory and vascular cytokines. tonometry, inflammatory, and vascular serum cytokines on treatment-naïve (n=20) and HIV uninfected (n=9) age matched control with metabolic profile, and Framingham score evaluation.

Study findings indicate that ART had a significant effect on reducing inflammatory and most vascular cytokines and arterial stiffness.

Comments:

As nearly half of participants receive and NNRT vs PI comparison of these 2 groups will be of interest as PI increased cholesterol and have a cumulative risk for CV events.

The difficulty of such studies with many markers only a few fit with the study hypothesis. Here the 2 inflammatory markers linked with non-AIDS events d-dimers and IL-6 were not different. Discussion on these 2 validated markers for CVV risks should be discussed in detail.

- In our population IL-6 has great variability and for this reason, has not been useful as a stable biomarker. Thus, we preferred to use hsCRP, which besides being a surrogate marker of IL-6 it is widely available in all laboratory settings in our country. Furthermore, a cut-off value for hsCRP is known for adverse cardiovascular outcomes in HIV. (http://www.biomedcentral.com/1471-2334/13/414)

LPS, sCD14 and beta-d-glucan play an important role in HIV-related inflammation and are linked with CV risk by imaging. Measurement of these markers will add value to the manuscript.

Isnard S, et al. Circulating β-d-Glucan as a Marker of Subclinical Coronary Plaque in Antiretroviral Therapy-Treated People With Human Immunodeficiency Virus. Open Forum Infect Dis. 2021 Mar 7;8(6):ofab109.

Isnard S, et al. Gut Leakage of Fungal-Related Products: Turning Up the Heat for HIV Infection. Front Immunol. 2021 Apr 12;12:656414.

In addition, the stress cytokine GDF-15 emerges as one of the best markers in CV disorders: ischemic conditions, atrial fibrillation and cardiac insufficiency and in 2022 in HIV. Assessment of one the best CV markers will be welcome for this study.

Royston L, . Growth differentiation factor-15 as a biomarker of atherosclerotic coronary plaque: Value in people living with and without HIV. Front Cardiovasc Med. 2022 Aug 26;9:964650.

-Great suggestion. However, sCD163 could act as a surrogate biomarker of CD14. sCD163 provides information on monocyte-macrophage activation. Both LDS and B-D-glucan have been indeed widely investigated but these markers were not included in the original study design. https://doi.org/10.3389/fimmu.2020.560381, https://pubmed.ncbi.nlm.nih.gov/25362192/

Discussion is much too long and should not be structured, no subchapters are necessary. Investigators should focus only on novelty as many data are confirming previous findings.

-We have eliminated the subchapters' headings and rewrote the discussion making it more concise and focusing more on the new findings. 

Increased in Cholesterol and TG are considered to augment as return to health at one year as generally patients gain 2 kg initially taken for HIV. Therefore, early changes are not linked to inflammation or PI effects on lipids after one year of ART.

Limitations: absence of cellular markers of adaptative and innate immunity.

-We agree and even though both cholesterol and TG had a statistically significant increase, the final post-ART mean levels were not clinically significant and did not require pharmacological treatment. 

As cells were not assessed a small paragraph on the role of adaptive immunity cells, such as CD4 cells are acknowledged to participate to CVD pathogenesis (Emeson et al., 1996; Zhou et al., 2005). 

CD4 cells were found infiltrated in atherosclerotic plaques (Saigusa et al., 2020). Recent evidence indicates that T helper 1 (Th1) cells have pro-atherogenic roles, whereas regulatory T cells (Tregs) can play a dual role being both anti-atherogenic or pro-atherogenic (George et al., 2012; Maganto-Garcia et al., 2011).Similarly in HIV, innate immunity: Elevated frequency of circulating non-classical monocytes (CD14dimCD16++) (Gu et al., 1998) and the intermediate CD14+CD16+ monocyte counts were associated to subclinical atherosclerosis (Hanna et al., 2017) and the expression of CX3CR1 on CD16+ monocytes predicted carotid artery thickness (Westhorpe et al., 2014)

-We acknowledge that CD4 do play a role in atherogenesis, and it will be taken into account in the next study we conduct. Nonetheless, given that arteriosclerosis is different from atherosclerosis and given the length of our discussion, we believe it would be more appropriate if we had measured atherosclerosis (e.g., intima media thickness).

Reviewer #2: This is important research. The study provides novel, prospective data in the field of HIV-associated CVD and warrants publication, even considering the relatively small number of participants. I do however think the manuscript could be improved before publication.

Major Comments:

My biggest concern is how the group of 20 HIV infected persons were recruited. The authors mention prior work with 51 individuals. Were these persons included from the prior cohort or are the participants all new? How were they selected if previously enrolled? Was everyone that was recruited followed up fully and included? Attrition? This needs to be discussed that possible bias may be evaluated. If they were chosen based on viral suppression alone, comparison with a virally unsuppressed group will be valuable and should be strongly considered.

All patients that attended the HIV Unit of the hospital were invited to participate in the study. In the beginning, we initially enrolled all 51 participants from our previous cross-sectional study. Unfortunately, due to the high drop-out rate (20%) and missing follow-up, we were only able to include 20 participants for the analysis from whom we had complete longitudinal data.

A significant limitation is the fact that the study essentially only included males. This should be discussed as a significant limitation as sex-differences were not evaluated and females are already significantly under-represented in CV research.

-Included in the limitations.

There were isolated delta (change over time) correlations mentioned , however, I miss a dedicated correlation analysis. Relevant correlations (or lack thereof) needs to be reported. Viral load, CD4 count, smoking, inflammatory markers, vascular markers and aortic stiffness?

-We believe that it was better to remove the isolated delta we reported. We think this could have been a type 1 error given the number of comparisons performed and the absence p-value adjustment. Moreover, given the small sample size, the risk of false negatives would be high to report an absence of association. We are looking to increase our sample size to determine the presence or absence of associations, especially with arterial stiffness.

However, after expressing our thoughts if both reviewers think a correlation analysis would be of benefit, we will happily add a correlation analysis section that could go as a Supplementary file. 

Varying ways of referring to HIV infected persons and the HIV cohorts are used in the manuscript. Decide on terminology and use this consistently.

-We decided to use the term PLH to refer to the HIV population in general.

NB: Mean CD4 count in table 1 is different to what is written in the abstract and the manuscript text.

-Corrected.

The discussion section is too long and reads as quite meandering. Every section should have a clear message and build on the message of the article. The final message of the article should be clear, and I think the authors should work on the conclusion (both in the abstract and the manuscript) to clearly summarise what they think their findings mean.

- We have substantially modified the discussion and did some changes to the conclusion.

Comments:

Introduction:

line 58-59: Unreferenced, inaccurate statement. CVD is not the leading cause of death in HIV. Rather say one of the leading causes of CVD.

-Corrected. It now reads “Cardiovascular complications are one of the leading causes of morbidity…”.

line 65 onward: Needs rewrite to clarify. There are sweeping statements without references and should be avoided. Smoking (as the authors mention later) has been associated with HIV in certain studies. This sentence reads as contradictory.

- Good observation. We removed that last sentence. We believe it is more important to emphasize that traditional risk calculators used in the general population may not be accurate in the HIV population.

line 68: reference please

- Reference added.

line 69 onward: Tonometry-based cfPWV is frequently employed in clinical research, but is not considered to be the definitive method of PWV measurement as it is implied in the text. The statement in the manuscript should be corrected to only read: 'cfPWV is considered the gold standard of aortic stiffness measurement.' Consensus has not been reached as to the definitive method to measure cfPWV and various techniques are well described.

read

Rajzer, Marek W; Wojciechowska, Wiktoria; Klocek, Marek; Palka, Ilona; Brzozowska-Kiszka, Małgorzata; Kawecka-Jaszcz, Kalina (2008). Comparison of aortic pulse wave velocity measured by three techniques: Complior, SphygmoCor and Arteriograph. Journal of Hypertension, 26(10), 2001–2007. doi:10.1097/hjh.0b013e32830a4a25

- Your comment is right. We modified it to state that carotid-femoral pulse wave velocity is the gold standard for aortic stiffness measurement without specifying the technique (i.e, oscillometric, mechanotransducer or tonometry).

line 77: Sparse work is available from low- and middle income countries, however, recent data have shown similar findings in sub-Saharan Africa. This may serve to strengthen your rationale for the research:

read

Robbertse PS, Doubell AF, Innes S, et al. Pulse wave velocity demonstrates increased aortic stiffness in newly diagnosed, antiretroviral naïve HIV infected adults: A case-control study. Medicine. 2022; 101:e29721.

- We added at the end of the sentence that there is limited data in developing countries.

line 113, 148 and other locations: Define all abbreviations with first use.

Those abbreviations and the rest have been defined.

line 150: p-value of <0.05

-Corrected

line 153-155: This does not seem correct. Paired samples t-test with 80% power to detect a 0.6m/s difference and the quoted SD with alpha=0.05, calculates to 29 pairs on SPSS. This is more than double the number quoted in the text.

Yes, you are correct. We forgot to mention that for this calculation it is necessary to include a correlation coefficient for paired data. Given that we were not able to find available such data in the literature, we used paired cfPWV from previous paired data in our lab in people living with HIV. We obtained a Pearson correlation coefficient of 0.81 and aimed to detect a 0.6 m/s difference and pooled 1.1 SD, as reported in our previous cross-sectional study between PLH and HIV(-). We used GPowet to do the sample size calculation.

Results:

Were any of the patients of vasoactive mediation? I think this is a worthwhile to mention, even if no-one was on these medicines.

- None of the study participants were on vasoactive medications. We added as part of the inclusion criteria for both PLWIH and controls.

I find the BMI findings the inverse of what I would expect when compared to the controls. Any ideas why this may be the case in your study?

- We believe that it could be an incidental finding due to the small sample size and/or even though they were treatment-naïve, 90% of HIV individuals had CD4 levels >200; thus, wasting syndrome was less likely in our sample. 

Not all abbreviations used in the tables are found in the legends. Please double check. Tables should be able to be able to be freestanding from the article and still be easy to interpret.

Abbreviations added to the table legends.

line 186: p=0.055 is not significant according to your study’s predefined level of significance. Rather state a trend toward significance for accuracy.

- It now reads: “c-LDL showed a trend towards a significant increase post-ART.”

line 205: Again, consistent use of wording to refer to the study groups. Post-ART should rather be ART-group or ART-experienced group.

-We changed it to ART-group.

line 208: ‘In the beginning’. Do the authors mean at baseline?

- Changed to “At baseline”.

Dedicated correlation analysis is an omission in my opinion as discussed earlier.

Discussion:

I would restructure the first paragraph to emphasise your own novel, longitudinal findings first. The mention of lack of longitudinal studies belong in the introduction. Rather state own strengths, than others weaknesses in this critical paragraph.

- The first paragraph was modified.

line 254: ‘slightly or did not decrease’. Reword this to communicate your findings clearly. Median hs-CRP essentially halved after ART. I would say something like ‘The hs-CRP showed a strong trend towards decrease, however, this did not reach statistical significance. hs-CRP, despite ART, remained significantly higher in ART-experienced persons when compared to the control group.’

- We rephrased that sentence.

line 265-270: This is all true. However, I think the reader would benefit from better packaging of these facts using our current theoretical framework of the factors that underpin cfPWV (especially in light of your young cohort, with likely negligible amounts of atherosclerotic disease). Pressure amplification by peripheral arterial tone. Vasomotor arterial tone is modulated by endothelial function, sympathetic tone, and the RAAS system. The manuscript would benefit from the incorporation of key concepts from the following articles (In general and in the setting of HIV). Use your excellent data to examine some of these concepts and what you think actually drive the increased (and decreased on ART) PWV mechanistically. Your study is explorative and should make the best use of your data.

read:

1. Cavalcante JL, Lima JAC, Redheuil A, et al. Aortic Stiffness. J Am Coll Cardiol. 2011;57:1511–22.

2. Robbertse PS, Doubell AF, Innes S, et al. Pulse wave velocity demonstrates increased aortic stiffness in newly diagnosed, antiretroviral naïve HIV infected adults: A case-control study. Medicine. 2022; 101:e29721.

- We rewrote the arterial stiffness paragraph to give a short introduction of arterial stiffness, how it is regulated and a possible mechanism that could explain our results from an inflammatory point of view.

line 319: reference

- We removed that part of the discussion. 

line 320: contrasting what? Use clear language.

- We modified this paragraph.

line 367-368: I would be careful with these strong statements. Your findings do not quite support this and I suggest rewording this statement. I would focus on what you could show: that was that ART had an overall positive effect (decreased vascular pathology markers, decreased markers of inflammation, and decreased cfPWV). The observation that aortic stiffness decreased with ART is important, as this means (as you stated) that a reversible component of aortic stiffness remains and there is likely a window before this becomes irreversible (collagen deposition, degradation of elastin etc). When this window is, remains unknown.

Furthermore, if cfPWV is used a surrogate of CV risk, you demonstrated increased risk compared to HIV uninfected persons at baseline. As cfPWV decreased at one year on ART (in a small group of virally supressed individuals), I would see this as a relative decrease in CV risk. Yet another reason to give ART. The residual risk despite ART is still of concern, and as you mentioned stratification may be an issue, as people on ART have higher CVD compared to those without HIV.

- We modified this last paragraph of the discussion. 

Limitations:

The sample is small and explorative and should be stated in no uncertain terms. Selection of the cohort needs to be more detailed, as there is a perceived risk of inclusion bias.

Typo’s, spelling, and language: Various errors present. I would suggest involving a proof-reader before re-submission.

line 30. PWHIV

-Corrected. We decided to use the abbreviation PLH across the manuscript.

line 36. young, treatment-naïve. Not treatment-naïve young.

-Corrected.

line 60: envelop

-Corrected. 

line 211: inflammation. rather biomarkers of inflammation

-Corrected. 

Minor comments for your discretion:

Title: I miss the word HIV, even though antiretrovirals imply this. I do not agree with the word "regression". Consider rewriting the title to refer to a decrease in aortic stiffness.

- We changed the title to: “Effect of antiretroviral therapy on decreasing of arterial stiffness, metabolic profile, vascular and systemic inflammatory cytokines in treatment-naïve HIV: A one-year prospective study.”

line 61: We generally refer to hs-CRP as high-sensitivity CRP, not highly sensitive CRP

- Corrected

line 62: remove increased

- Removed. 

line 64: non-related should be unrelated

-Modified that last sentence.

---

## [Decision Letter · Decision Letter 1]

13 Feb 2023

PONE-D-22-26706R1Effect of antiretroviral therapy on decreasing of arterial stiffness,  metabolic profile, vascular and systemic inflammatory cytokines in treatment-naïve HIV: A one-year prospective study.PLOS ONE

Dear Dr. Andrade,

Thank you for submitting your manuscript to PLOS ONE. After careful consideration, we feel that it has merit but does not fully meet PLOS ONE’s publication criteria as it currently stands. Therefore, we invite you to submit a revised version of the manuscript that addresses the points raised during the review process.

ACADEMIC EDITOR: see comments below

Please submit your revised manuscript by 1 week. If you will need more time than this to complete your revisions, please reply to this message or contact the journal office at plosone@plos.org. Please include the following items when submitting your revised manuscript:A rebuttal letter that responds to each point raised by the academic editor and reviewer(s). You should upload this letter as a separate file labeled 'Response to Reviewers'.A marked-up copy of your manuscript that highlights changes made to the original version. You should upload this as a separate file labeled 'Revised Manuscript with Track Changes'.An unmarked version of your revised paper without tracked changes. You should upload this as a separate file labeled 'Manuscript'.If applicable, we recommend that you deposit your laboratory protocols in protocols.io to enhance the reproducibility of your results. Protocols.io assigns your protocol its own identifier (DOI) so that it can be cited independently in the future. For instructions see: https://journals.plos.org/plosone/s/submission-guidelines#loc-laboratory-protocols. Additionally, PLOS ONE offers an option for publishing peer-reviewed Lab Protocol articles, which describe protocols hosted on protocols.io. Read more information on sharing protocols at https://plos.org/protocols?utm_medium=editorial-email&utm_source=authorletters&utm_campaign=protocols.

We look forward to receiving your revised manuscript.

Kind regards,

Eliseo A Eugenin, Ph.D.

Academic Editor

PLOS ONE

Journal Requirements:

Additional Editor Comments (if provided):

Dear Dr. Andrade

Thank you for submit your manuscript to PLOSone. Please add the excellent comments of reviewer 2 and send the manuscript back

Eliseo

Reviewers' comments:

Reviewer's Responses to Questions

**Comments to the Author**

1. If the authors have adequately addressed your comments raised in a previous round of review and you feel that this manuscript is now acceptable for publication, you may indicate that here to bypass the “Comments to the Author” section, enter your conflict of interest statement in the “Confidential to Editor” section, and submit your "Accept" recommendation.

Reviewer #1: All comments have been addressed

Reviewer #2: (No Response)

2. Is the manuscript technically sound, and do the data support the conclusions?

Reviewer #1: Yes

Reviewer #2: Yes

3. Has the statistical analysis been performed appropriately and rigorously? 

Reviewer #1: Yes

Reviewer #2: Yes

4. Have the authors made all data underlying the findings in their manuscript fully available?

Reviewer #1: Yes

Reviewer #2: Yes

5. Is the manuscript presented in an intelligible fashion and written in standard English?

Reviewer #1: Yes

Reviewer #2: Yes

6. Review Comments to the Author

Reviewer #1: issues have been addressed including statistics and discussion conclusion.

Prospective data with relatively advanced patients add to current knowledge

Reviewer #2: Thank you for the author's responses to my comments. I am mostly satisfied with the responses and the amendments to the manuscript. The new title is excellent.

I discuss a few comments that still need to be adequately addressed below:

Original comment 1: Recruitment

Thank you for clarifying. Selection bias is be a factor that needs consideration by the reader and merits careful explanation. If the original cohort had 51 participants, the completion rate is 39%. (higher attrition than most HIV research, granted that PLWH are known to have high attrition). The quoted 20% dropout rate is therefore confusing, as the numbers do not add up. Please state how many were lost, withdrew, died, or had incomplete records. I would state this high attrition rate as a limitation of the study - it's higher than to be expected and likely influenced your results.

Original comment 3: Correlation analysis

I see the merit in the argument and agree that a future, larger sample size would be more appropriate for this. Please state explicitly in the methodology that a correlation analysis was not undertaken due to the small sample size in this explorative work.

New comments:

1. Limitations section: "Despite that we aimed to isolate the effect of HIV on arterial stiffness by choosing PLH without any other comorbidities and comparing them with healthy controls, there may be a series of genetic, lifestyle..."

Please improve the language here: My suggestion: "Despite our aim to isolate the effect HIV on arterial stiffness by choosing PLH without any other comorbidities and comparing them with healthy controls, there may be ..."

2. There are new, untracked additions to the manuscript. All changes should be stated or better yet, clearly tracked/highlighted for the editor.

Of note: "The literature vastly describes that ART does not completely suppress viral load, particularly in the viral reservoirs".

This is not in the academic style of writing. Rephrase. I suggest: "It is well described that ART does not completely suppress viral load..."

3. Define all abbreviations with first use: hs-CRP is never defined. hsCRP vs hs-CRP are both used in the manuscript.

4. I still feel the article could receive additional language manicuring. An academic proof reader should be considered.

I look forward to your work in print.

Regards,

7. PLOS authors have the option to publish the peer review history of their article (what does this mean?). If published, this will include your full peer review and any attached files.

Reviewer #1: **Yes: **Jean-Pierre Routy

Reviewer #2: No

---

## [Author Response · Author response to Decision Letter 1]

20 Feb 2023

Original comment 1: Recruitment

Thank you for clarifying. Selection bias is be a factor that needs consideration by the reader and merits careful explanation. If the original cohort had 51 participants, the completion rate is 39%. (higher attrition than most HIV research, granted that PLWH are known to have high attrition). The quoted 20% dropout rate is therefore confusing, as the numbers do not add up. Please state how many were lost, withdrew, died, or had incomplete records. I would state this high attrition rate as a limitation of the study - it's higher than to be expected and likely influenced your results.

- Thank you for your comment. Indeed, it is very important that we explain and make sure there was no selection bias. As stated in the revised manuscript, we selected the first 20 patients, with no selection of any kind, who had completed one year of viral suppression and had complete baseline and final data. We compared 31 patients from our cohort that were not included in the analysis to ensure that there were no differences in clinical or demographic characteristics.

Original comment 3: Correlation analysis

I see the merit in the argument and agree that a future, larger sample size would be more appropriate for this. Please state explicitly in the methodology that a correlation analysis was not undertaken due to the small sample size in this explorative work.

-Included in the methods section.

New comments:

1. Limitations section: "Despite that we aimed to isolate the effect of HIV on arterial stiffness by choosing PLH without any other comorbidities and comparing them with healthy controls, there may be a series of genetic, lifestyle..."

Please improve the language here: My suggestion: "Despite our aim to isolate the effect HIV on arterial stiffness by choosing PLH without any other comorbidities and comparing them with healthy controls, there may be ..."

- Changed as suggested.

2. There are new, untracked additions to the manuscript. All changes should be stated or better yet, clearly tracked/highlighted for the editor.

Of note: "The literature vastly describes that ART does not completely suppress viral load, particularly in the viral reservoirs".

This is not in the academic style of writing. Rephrase. I suggest: "It is well described that ART does not completely suppress viral load..."

- Changed as suggested by the reviewer.

3. Define all abbreviations with first use: hs-CRP is never defined. hsCRP vs hs-CRP are both used in the manuscript.

- All hs-CRP abbreviations are now consistent.

4. I still feel the article could receive additional language manicuring. An academic proof reader should be considered.

- We have several minor corrections to the grammar.

---

## [Editor Report · Decision Letter 2]

22 Feb 2023

Effect of antiretroviral therapy on decreasing arterial stiffness,  metabolic profile, vascular and systemic inflammatory cytokines in treatment-naïve HIV: A one-year prospective study.

PONE-D-22-26706R2

Dear Dr. Andrade,

We’re pleased to inform you that your manuscript has been judged scientifically suitable for publication and will be formally accepted for publication once it meets all outstanding technical requirements.

Kind regards,

Eliseo A Eugenin, Ph.D.

Academic Editor

PLOS ONE

Additional Editor Comments (optional):

Dear Dr. Andrade

Thank you for submitting your manuscript to PLOSone and include the changes requested

Eliseo Eugenin
---

## [Editor Report · Acceptance letter]

10 Mar 2023

PONE-D-22-26706R2 

Effect of antiretroviral therapy on decreasing arterial stiffness,  metabolic profile, vascular and systemic inflammatory cytokines in treatment-naïve HIV: A one-year prospective study. 

Dear Dr. Andrade-Villanueva:

I'm pleased to inform you that your manuscript has been deemed suitable for publication in PLOS ONE. Congratulations! Your manuscript is now with our production department. 

Kind regards, 

on behalf of

Dr. Eliseo A Eugenin 

Academic Editor

PLOS ONE